# Design and Implementation of OPC UA-Based VR/AR Collaboration Model Using CPS Server for VR Engineering Process

**Jeehyeong Kim** and **Jongpil Jeong** *

Department of Smart Factory Convergence, Sungkyunkwan University, 2066 Seobu-ro, Jangan-gu, Suwon 16419, Korea; ghyeong2@naver.com
* Correspondence: jpjeong@skku.edu; Tel.: +82-31-299-4267

**Abstract:** In order to cope with the changing era of the innovative management paradigm of the manufacturing industry, it is necessary to advance the construction of smart factories in the domestic manufacturing industry, and in particular, the 3D design and manufacturing content sector is highly growthable. In particular, the core technologies that enable digital transformation VR (Virtual Reality)/AR (Augmented Reality) technologies have developed rapidly in recent years, but have not yet achieved any particular results in industrial engineering. In the manufacturing industry, digital threads and collaboration systems are needed to reduce design costs that change over and over again due to the inability to respond to various problems and demands that should be considered when designing products. To this end, we propose a VR/AR collaboration model that increases efficiency of manufacturing environments such as inspection and maintenance as well as design simultaneously with participants through 3D rendering virtualization of facilities or robot 3D designs in VR/AR. We implemented converting programs and middleware CPS (Cyber Physical System) servers that convert to BOM (Bill of Material)-based 3D graphics models and CPS models to test the accuracy of data and optimization of 3D modeling and study their performance through robotic arms in real factories.

**Keywords:** virtual reality; augmented reality; cyber physical system; OPC UA; CAD





## 1. Introduction

Recently, innovations in the ICT (Information and Communications Technology) field have begun to rapidly change existing manufacturing methods into smart manufacturing methods or CPS (Cyber Physical Systems) [1]. These manufacturing methods can use IoT (Internet of Things) technologies and web-based services to communicate and interact with other products in a factory environment. Due to the COVID-19 pandemic, many manufacturers have come to seriously consider the problem of flexibly adjusting production according to environmental changes while promoting employee safety. Depending on the situation regarding changes in infectious diseases, the demand for certain products varies substantially by country or region. Specifically, manufacturing companies that rely on small-scale production of multiple varieties should come up with countermeasures such as quickly readjusting their production lines and educating their employees on new production manuals in line with changes in consumer demand. As telecommuting continues to increase, the need to clearly check the status of production plants online is increasing as well. For manufacturers to overcome the COVID-19 pandemic crisis, it has been stated that it is necessary to actively introduce robot automation that can promote the safety of employees and increase production efficiency, and to combine virtual convergence reality technologies such as Digital Twin and VR (Virtual Reality)/AR (Augmented Reality) technology.

Although the development of Digital Twin started from the Aerospace industry, the industry which is exploring the technology the most is the manufacturing indus-

try. Digital Twins have been described as the key enablers of Industry 4.0 and Smart Manufacturing [2,3]. Any manufactured product goes through four main phases throughout its life cycle: design, manufacture, operation, and disposal. Smart manufacturers can leverage Digital Twins in all four phases of the product [4,5]. During the design phase, Digital Twin allows the designers to verify their product design virtually, which enables them to test different iterations of the product and choose the best one. An example of this is the car manufacturer Maserati, which used Digital Twin for optimizing car body aerodynamics using wind tunnel tests virtually, which are elaborate and expensive otherwise. Unlike traditional bike designing methods, which are based on the designers' knowledge and experience, the Digital Twin of a bicycle constantly collects the data from the physical space, which could be compared, analyzed, and used for designing or redesigning next-generation bicycles. With customer reviews and usage habits, designers can get a better understanding of customer requirements, which can be translated into better and improved features. Capturing customer preferences via a Digital Twin lets businesses know about the market trends which can be integrated with the customer usage data to see the effects on product performance. This allows businesses to take design-decisions and incorporate them in an informed way, thus making the process of integrating the customer feedback into the product to deliver customized products easier [6].

One of the most developed directions in manufacturing, especially in the scope of Industry 4.0, is the robotization and automatization of the production lines [7]. Digital Twin is playing a crucial role in this integration, as industrial robots are being programmed with mainly three methods, which are closely related to the twinning of the manufacturing equipment. Those three methods are: (i) Offline, a dedicated virtual environment (usually each robot brand has their own) for programming each aspect of the robotic cell for later deployment through the network to the physical robot; (ii) Online, which is being adapted by means of sensor information, usually being twinned in the dedicated virtual environment (e.g., ROS (Robot Operation System), and is able to directly affect the pre-programmed path and routine of the robotic systems; (iii) Manual, which is robot programming by the usage of a flex pendant, but with the introduction of VR/AR interfaces, it also uses a twin for manipulation near the virtual robot remotely [8]. As can be observed, combinations of those methods exploit the virtual twin of the robotic cell and are widely used across the various industrial sectors. Moreover, Digital Twin is used as a validation tool for HRC (Human–Robot Collaboration) safety standards, to evaluate the safety level of the system first, for example, using VR human avatars, before experiments with real operators in the actual system.

In this study, it is possible to visually check the converted 3D model in a virtual environment for product designers, purchasing teams, and laboratory staff of small and medium-sized enterprises that lack manpower and capital as well as large enterprises. Then, one develops a VR/AR-based product inspection environment that enables collaboration, which is the main goal in terms of the CAD data conversion process of the CPS-based VR/AR system. In addition, the research is focused on establishing a cooperative relationship between each system entity that observes, calculates, and manipulates physical phenomena through CPS.

The smart factory market is expanding with the development of technology following the fourth industrial revolution. Advancements in smart factories include factory automation, and various mechanical devices perform the process automatically. With the introduction of smart factories, field workers mainly carry out inspections and fault repairs, and field workers need a lot of prior learning to check and repair various mechanical devices. In the manufacturing industry, skilled workers are aging, and it is difficult to properly train unskilled workers due to the increased complexity of manufacturing facilities. Collaboration ideas are also emerging as the need for untact emerges due to the COVID-19 pandemic. In the manufacturing industry, collaborations often use AR and VR technologies, and as a result, related jobs are increasing worldwide.

To cope with the changing era of the fourth industrial revolution's innovative management paradigm, it is necessary to accelerate the construction of smart factories in the manufacturing industry, and the 3D design and manufacturing content sectors show high potential for growth. It is also necessary to introduce a standardized CPS communication platform technology to facilitate the construction of smart factories in the manufacturing industry [9]. OPC UA (Open Platform Communications Unified Architecture) has become an integral part of the construction of CPS communication platforms that must have an independent and standardized communication method between different species. OPC UA technology is considered to be important and valuable for data exchange and standardized data exchange for various industrial applications such as ERP (Enterprise Resource Planning)/MES (Manufacturing Execution System)/PDM (Product Data Management). To increase the reliability of BigData, OPC UA, a device-friendly communication, high value-added is created with the high reliability of simulation. OPC UA has real-time functionality in Ethernet environments with TSN (Time Sensitive Network), thus making it suitable for the high performance needs of the VR/AR network, and OPC UA-based CPS connects closed networks as standard networks in response to numerous industrial protocols to integrate OT and IT. Further, applications that require standardized data integration, such as ERP/MES, can search and collect data without having to install a separate network.

This study proposes a VR/AR collaboration design environment based on CPS communication platform technology and quality function development. This study makes the following contributions:

- We extract VR/AR rendering and data of 3D design files through data exchange and distribution through CPS servers with OPC UA and propose a model specialized in manufacturing collaboration.
- While referring to the OPC UA specification of international industry standards, the system is constructed by connecting both OPC UA modeling and CPS Node modeling with technical convergence with CPS.
- The CAD Data Import Interface allows designers to insert CAD data into the CPS and reads the CAD data interface to a VR Parser and converts it into a 3D object.
- BOM (Bill of Material) is extracted from CAD or migrated from legacy DB to refer the contents of the BOM to the related node in the Address Space of the CPS server to enhance the functionality of the physical engine based on the BOM.

The structure of this paper explains the necessity of collaboration in Industry 4.0 and the necessity of this study in Section 1. In Section 2, we examine the related studies of OPC UA-based CPS and VR/AR systems. In Section 3, we describe the composition and role of the VR/AR collaboration model and VR engineering process architecture proposed in this paper. In Section 4, we measure the address space conversion accuracy and optimization of the BOM of the actual factory equipment, and measure the rendering speed. Section 5 summarizes the proposed architecture, implementation, and test results.

## 2. Related Work

In this study, it is necessary to first understand the characteristic CPS factors and clarify the design review. Next, you need to understand VR/AR.

At the core of a CPS is the physical part based on mechatronic products [10] or intelligent mechatronic products. Such a system gets a label called CPS, which has the ability to communicate with other products [11]. The most important features of CPS are high autonomy, network communication, personalization capabilities, and general user affinity [12]. Moreover, CPS features dynamic reconfigurability over the entire product life cycle and real-time responsiveness to environmental changes. Initiatives such as German RAMI 4.0 (Reference Architectural Model for Industrie 4.0) [13] or US IIRA (Industrial Internet Reference Architecture) [14] aim to provide guidance for building IoX (Internet of Everything) devices for industrial use cases [15]. Each component within the CPS has a so-called management shell that associates components with other components of the reference architecture. The intended benefits of this linkage result in overall cost savings due

to improved planning processes or utilization of feedback information from real CPS for identifying bottlenecks as well as future process and product optimization [16]. We describe ontology-based concepts with the aim of obtaining and providing knowledge feedback on product design and deriving rules for fault prevention [17]. Thus, design reviews of CPS can access on-site reports from previous generations of physical components as well as feedback data and services provided by product types and components. In this work, we will use an approach using CPS based on OPC UA to improve the design review process of CPS [18–23].

In the process of the switch to CPS, more data will be generated in the future, and this will help improve the physical environment. To bridge the gap between the physical and digital/virtual worlds, VR is a suitable technology for visualizing and interacting with geometric 3D models in virtual environments [24]. In general, when VR is used for design review and other engineering tasks, the goal is to achieve cost savings and improve or accelerate related processes. There are many approaches in the academic literature that aim to support multiple engineering applications through VR. VR makes it easy to verify design, and manufacturing costs can be reduced by replacing review processes involving physical prototypes and mock-ups with processes involving virtual prototypes [25]. Further, the product modifications can be implemented quicker if physical parts have not already been used. In addition to its general feasibility, it focuses on improving user interaction within a VR environment. Mohamed Tabaa et al. [26] have developed a hardware and software-based user interface that simplifies the interaction of VR review processes. Adrian Harwood et al. [27] investigated tools and methods for the CAD-like interaction and manipulation capabilities of 3D models. Research on VR-supported design reviews shows that participants are more likely to detect errors in 3D models within VR scenes than traditional CAD software approaches on PC screens. Further, conducting design reviews and interacting with 3D models is more intuitive and natural for non-CAD professionals due to their high immersion. In this work, we consider the design review of CPS and therefore connect the digital and physical worlds. In particular, the focus on the proposed implementation depends on the availability of accessing and interacting with digital content as well as representing object information in a collaborative and immersive environment.

*2.1. OPC UA-Based CPS*

OPC UA-based CPS presents the architecture of a Smart Factory CPS using the structure and concepts of the OPC UA Framework [28], which is an international smart factory standard, and it relates to the structure and method of dynamically constructing CPS models with OPC UA modeling.

OPC UA-based CPS is largely functionally composed of OPC UA Server and Factory CPS Model [29]. The CPS Server consists of an OPC CPS Node Generator, which automatically creates a systematic connection between the Factory CPS Model and the OPC UA Client and a variety of OPC FileA sensors and clients, along with an information exchange service for data exchange between CPS Connect OPC UA and external systems. The OPC UA Address Space is a variable system that changes the value of the modeled entire object node in real time. The OPC UA specification manages and delivers all values through the Address Space. Within the Centralized CPS Server, OPC UA Pub/Sub, OPC UA Monitored Item, Alarm/Event, and Historian, all data are exchanged and updated through the OPC UA Address Space.

The Information Exchange Service is responsible for exchanging and delivering data with external systems. It consists of OPC UA standard specifications and asynchronously distributes data to external brokers and clouds through OPC UA Pub/Sub. The OPC UA Monitored Item periodically registers the node value that the system and application that wants to exchange data from the outside and accordingly transmits data periodically. When registering with an OPC UA Monitored Item, attributes such as transmission period and data transfer filtering are registered along with the index value of the node to be obtained, and the OPC UA Monitored Item transmits the data for each session accordingly [30]. OPC

UA Alarm/Event distributes event data by simultaneously generating the alarm/Event internally and externally whenever that event occurs, in accordance with the attributes of the Alarm/Event defined in OPC UA Node Modeling [31]. The CPS Connect OPC UA is responsible for updating and changing all data in the OPC UA Address Space and for transferring internal alarms/events from the Information Exchange Service to CPS Node Control, CPS Logic Control, and Product Process Control inside the Factory CPS Model. The OPC CPS Node Generator automatically generates the properties and codes required for CPS Node Control, CPS Logic Control, and Product Process Control, all of which are registered in OPC UA Address Space and OPC UA Client by parsing OPC UA Modeling [32]. The OPC UA Historian records all updated information in the Address Space. The recording method can be used by selecting a relational database, a NoSQL database, an XML, a binary file, etc.

The OPC UA-based CPS architecture allows CPS to be configured with only OPC UA Modeling and continuous system operation without the need for a system redesign or program changes or downtime [33]. By embracing the international standard OPC UA Spec, numerous facilities, machinery, and robot products can be accommodated in a single protocol. Managing data in real time through Address Space increases the efficiency of horizontal data distribution and exchange operations, significantly reduces work time and costs, and solves the problems of data loss, system performance degradation, and real-time processing delay.

### 2.2. Industrial VR Technology

VR is one of the key technologies of digital transformation. The powerful and immersive hardware and software systems that are currently on the market allow one to visualize complex systems used in technology, nature, physics, chemistry, and anatomy in a realistic virtual environment. This allows us to fully immerse ourselves in the observed virtual environment and interact with graphic objects and systems, thus creating new possibilities for describing and communicating complex and abstract system behavior to non-experts in a concrete and understandable manner. VR technology has yet to have a breakthrough technology that can be recognized in the industry, except for large companies in the automobile and aircraft industries, and small and medium-sized enterprises are particularly facing difficulties building VR systems and applications, introducing them into business processes, and operating them economically. This is mainly due to the lack of standardized interfaces between CAD systems and VR systems and the substantial difficulty involved in implementing VR applications. At present, this data integration remains largely divided, especially when there is a need for more than just geometry to be transferred outside the software environment of CAD system providers, and the "Industrial Utilization of Virtual Reality in Product Design and Manufacturing" survey [34] suggests one of the key issues in VR. CAD data and related design data in the product design process cannot be directly sent to VR applications, so basic CAD data must be converted to VR data. For companies characterized by frequent and rapid CAD data changes in development processes, use of VR technology is currently difficult and almost unimaginable in terms of efficiency, time, and economy. To utilize VR potential in engineering and industrial environments, there is an urgent need for solutions that promote the implementation and setup of VR applications—particularly for small and medium-sized enterprises—and enable intelligent use of this technology. In addition, AR technologies, including object recognition, deep point cloud scanning, gesture manipulation, and VR have gradually been introduced in recent years, and technologies related to the fusion of VR and AR have been explored [35].

### 2.3. CAD Data Conversion Process for VR/AR Systems

VR refers to an artificial environment that feels like reality. It also refers to a computer graphic system technology that allows a user to immerse himself in a 3D virtual environment using interface equipment, and the core of VR is the concept of realism, which allows users to exist in a space or virtual place.

AR is a technology that superimposes a real image or a three-dimensional virtual image of a background and displays it as a single image, and this is called Augmented Reality. At present, the main method is one that involves receiving a real-time image through an image input device mounted on a mobile device, then overlapping, displaying, and providing information on a 3D object.

The advantage of VR is that it can give a direct feeling of space to a space that does not actually exist, so it is easy to grasp the situation without going directly to the site. Further, visual elements have the advantage of being able to be observed and felt more directly than other senses, so they can be accurately adapted and highly understood, and they can thus be used conveniently by users. It is also possible to prevent problems caused by construction errors in actual sites by simulating them virtually through collaboration, and it has the advantage of reducing costs such as labor costs by reducing redesign.

The business processes of most industrial companies that develop and manufacture technology systems and products are CAD-based [36–40]. CAD models are the primary targets for recording and representing technical product data over the entire product lifecycle [41]. All lifecycle processes and tasks are based on CAD models to create, process, and store design, simulation, manufacturing, production, logistics, training, and maintenance data. Many standardized SETP and IGES data formats have been defined for this purpose specifically. These standardized data formats are handled by the most common engineering tools over the entire product lifecycle, except for VR engines. Over the past years, the development and application of VR technology has mainly been driven by the gaming and entertainment industries. Graphical requirements in these areas are completely different from those in the engineering domain. For example, the data content of a game object is very superficial compared to that of an engineering object. For game objects, the surface shape is important for VR applications, but for engineering VR applications, complete metadata, including material and tolerance information for parts, is needed. This has led to common VR engines and VR data interfaces and processors that can be implemented for game design to focus on processing only minor surface data and reducing metadata. To solve this problem, it is necessary to automatically process the data transfer process from the CAD environment to the VR environment.

Numerous studies have been conducted over the past few years with the goal of addressing this CAD-VR data conversion problem and reducing the gap between CAD and VR systems [34]. The focus is on transmitting CAD data to VR using a PDM [37]. CAD-VR data transformation workflows were defined using the VRML (Virtual Reality Modeling Language) standard [42], and approaches for VR-based collaborative platforms were defined using the STEP (Standard for the Exchange of Product Model Data) and JT (Jupiter Tesla) standards [43]. However, this approach focuses on the data transformation of a particular CAD system that uses a particular CAD data format [36,44,45]. Specifically, it only considers VR applications [46] and data used in VR-Cave, and it does not apply to HMD (Head Mounted Display)-based VR solutions.

## 3. OPC UA-Based VR/AR Collaboration Model

This research aims to reduce design costs, which have changed often due to various problems and the demands of manufacturing products. This study provides participants with VR/AR with 3D rendering virtualization to improve design efficiency. Through data exchange and distribution through CPS servers applying OPC UA, an international standard technology, VR/AR rendering and data of 3D design files are extracted and a platform specialized for manufacturing collaboration is developed. This provides an editor that allows users to write AR content themselves by utilizing VR modeling, and it supports re-reflection of product operation and operation results through simulators in design and rapid rendering through a central virtual rendering server. From design to future maintenance, it provides collaboration functions over the entire life of the product, holds data, and enables AR inspection and maintenance using 3D rendering models to

collect and analyze data that is difficult to obtain using existing research methods through VR to visualize BI (Intelligence Business).

### 3.1. VR/AR Collaboration Model

Figure 1 depicts the main components of the VR/AR collaboration model using CPS by dividing the VR/AR layer, Visualization layer, and CPS layer.

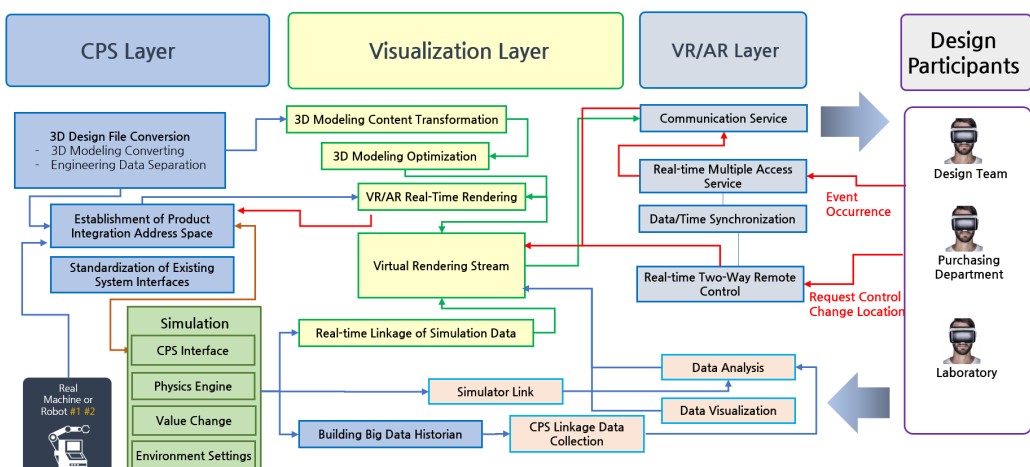

**Figure 1.** VR/AR collaboration model construction process.

The CPS Layer is the central source data store of this system and serves as middleware. The standard of OPC UA, an international standard technology for smart factories used for Digital Twin technology in factories, was applied here. OPC UA is not just a protocol, but it standardizes all the authentication, exchange, storage, alarms, and distribution required for data exchange. The CPS service has a real-time memory structure called Address Space, which serves the machine's information modeling and attribute values in real time. In this system, real-time data generated by a simulator or an actual machine is delivered to a VR/AR rendering service. It also exchanges data with standard specifications on all existing systems that require data interworking and exchange. In big data analysis, all information data generated in this cooperative design is stored and used to generate big data. The simulation, which is characteristic of this service, calculates real-time data, such as the actual machine operating based on the physical engine, stores and analyzes it, and lastly returns the analysis results to the simulator to improve the accuracy and virtuality of the simulation. Further, by visualizing and serving all data, participating designers provide services that enhance design capabilities through visualized data analysis. Moreover, by providing a management module, it offers systematic services such as all of the user management, security management, authentication management, access management, and history management services necessary to service this 3D collaboration design system.

The visualization layer optimizes 3D modeling to convert models extracted from 3D design files into VR/AR, converts them into stream images from virtual rendering servers, and simultaneously transmits simulator values to VR/AR in real-time interworking with CPS. It synchronizes the time and data of simultaneous multi-access users; detects the location, control, and event occurrence of design participants; and communicates in both directions with the VR/AR rendering server in real time to service the video stream without delay. Further, the voices of participating designers are synchronized in real time through VOIP.

The VR/AR Layer based on 3D design modeling and physical engine is used for product development, and the contents necessary for product inspection are developed and applied to deliver inspection results to CPS systems. The CPS system delivers it to the existing system required for existing inspection and quality. Based on the 3D design modeling and physical engine used for product development, it provides contents and

services that allow customers to perform maintenance and defect repair through an AR service that allows them to watch video transmitted through AR goggles and Live Cam.

Figure 2 shows the system configuration diagram. It is a structure in which several engines and servers such as real-time rendering and big data analysis engines are combined with existing CPS servers and VR/AR collaboration servers. Digital Twin is not essential for untact manufacturing VR/AR collaborative AI systems, but the most effective configuration is to build real-time Digital Twins for factory machines and robots. In the OPC UA-based CPS server and VR/AR collaboration system, the real-time data of a Digital Twin is an important system for data analysis and prediction. Collaborators and fielders are connected to each other with videos through communication, and the fielders' images are shared with collaborators. If a Digital Twin is used, VR models and CPS data are overlaid over real-time images with an AR function.

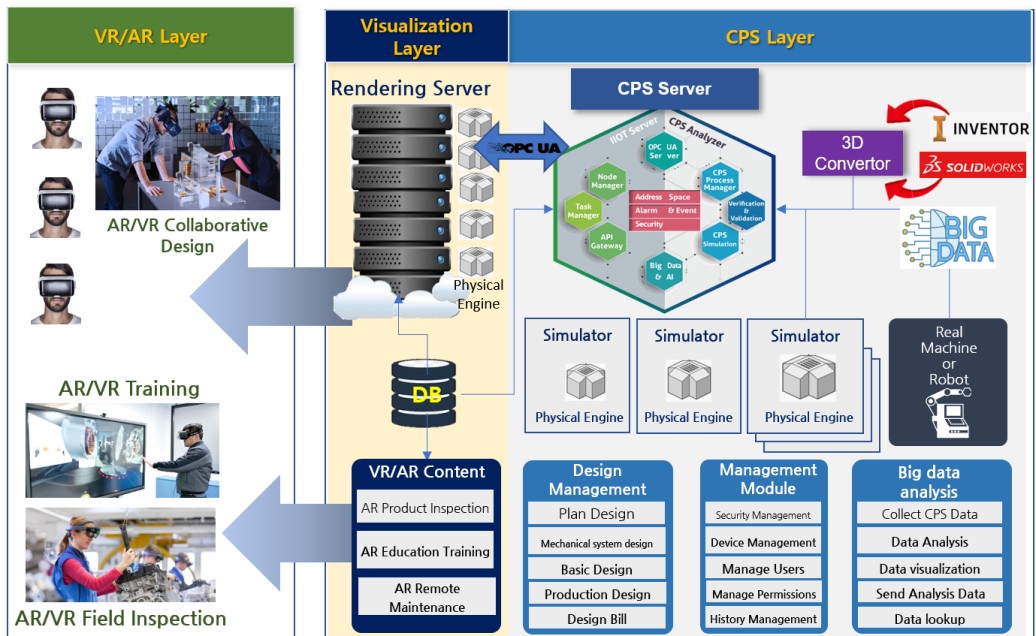

**Figure 2.** VR/AR collaborative model.

### 3.2. VR Engineering Process

The system maps the real-time values of the existing CPS server Address Space to the AR Object in the manner shown in Figure 3. The existing collaboration system has focused on object recognition and VR matching using VR/AR technology alone, and it only raises the actual value of AR through some data interfaces. Overseas, only PTC products receive actual data through Kepware's CPS Server. Figure 3 shows that rendering is possible through 3D design files by exchanging data through the CPS server to which OPC UA, an international standard technology, is applied. CPS maps from OPC UA Server to Address Space to exchange equipment information in real time without delay. As a central address space, the CPS system is built from multiple OPC UA Servers in the field as one huge central address space, and with one connection, the entire factory information can be mapped to the product inspection system in real time without delay.

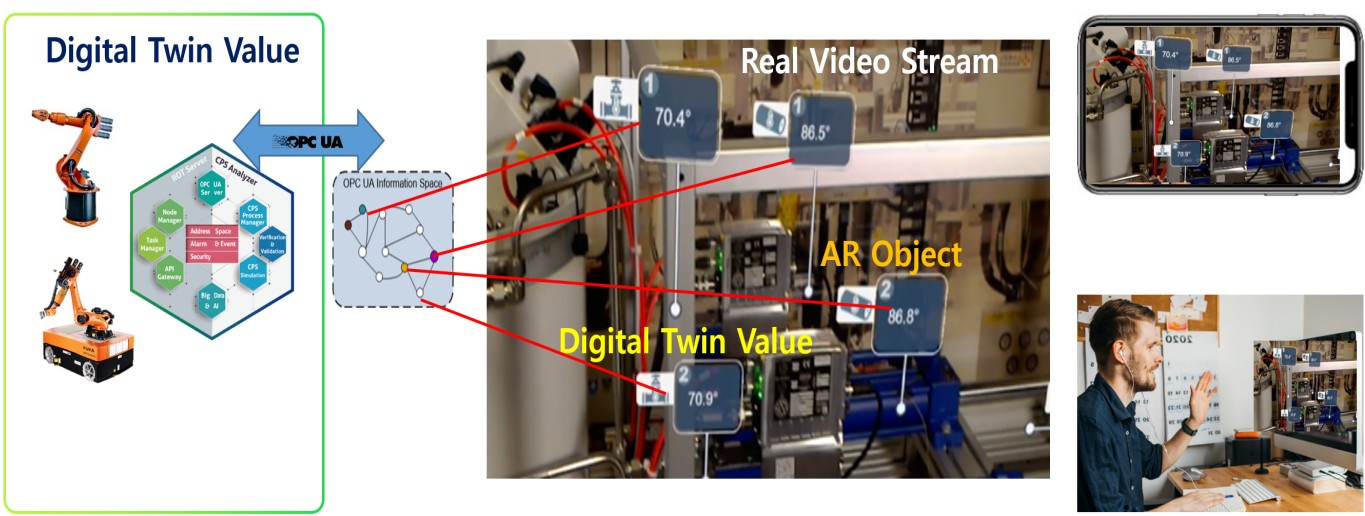

**Figure 3.** OPC UA information space of VR engineering.

The existing CPS played the role of collecting data from all equipment in the plant. In addition to the existing role, the proposed CPS plays a role of linking data to the 3D model through the OPC UA protocol. CPS performs the role of middleware including object information and value of the 3D model object delivered to the rendering server by applying technology based on the OPC UA Framework, which is an international table. The data stored in the CPS is utilized not only in the 3D model but also in the rendering server. In addition to communication, with computational control as the core concept, the physical world objects that coexist with humans and objects such as sensors, actuators, and embedded systems are fused into the cyber world, and the data communicated with the simulator are calculated and the results are redistributed. Using the OPC UA specification of international industrial standards, the system is constructed as depicted in Figure 4, with the smart factory optimized and both OPC UA modeling and CPS Node modeling connected with technological convergence with CPS. It converts CAD 3D binary files such as Solidworks into OPC UA Address Space data. If Edge is needed, data are collected from simulators, real machines, robots, etc., and data are delivered to CPS.

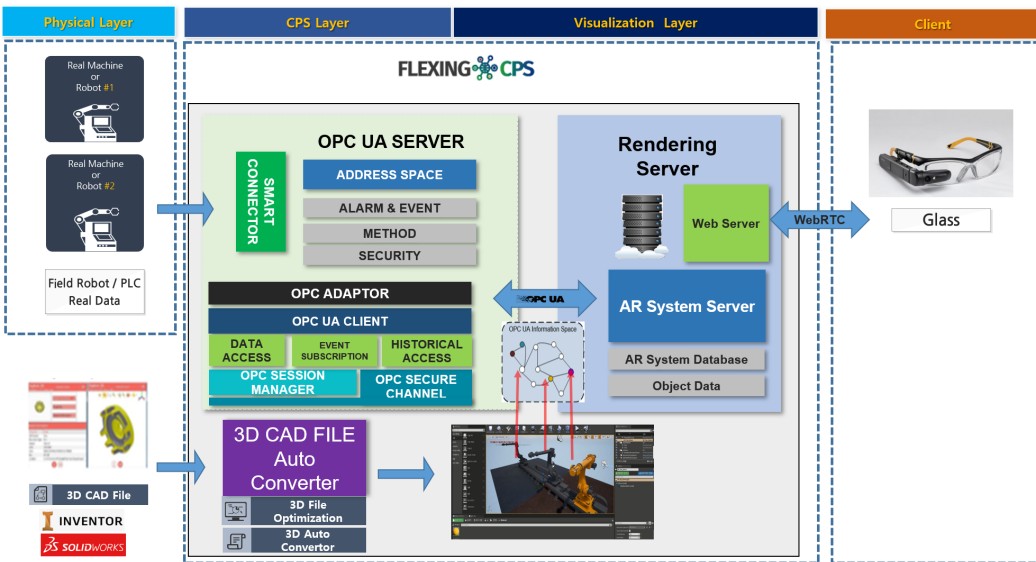

**Figure 4.** System architecture of VR engineering.

The design file is treated to the modeling and separation conversion of the design file, and the process proceeds as depicted in Figure 5. The CAD Data Import Interface provides an interface that allows designers to insert CAD data into applications and reads CAD Data Interface to VR Parser and converts it into 3D objects. For visualization, if a designer places 3D objects to collaborate and then performs virtual collaboration space work, CAD models are input to create virtual space objects through a 3D Converter, and designers can freely assemble and decompose objects in virtual space. It is also possible to easily create a virtual collaboration space so that it can be displayed to collaborators.

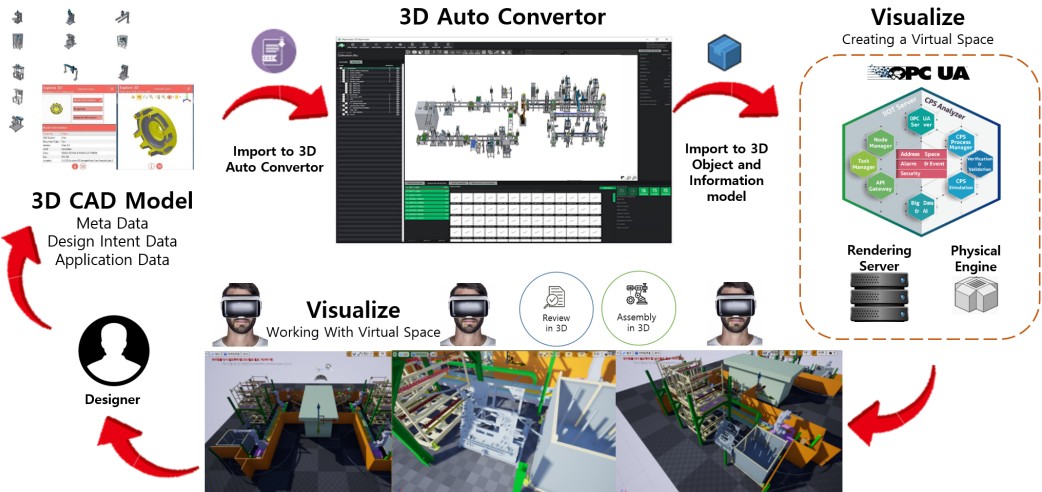

**Figure 5.** Model separation from 3D design files.

Further, as shown in Figure 6, the contents of the BOM are referred to the relevant Node in the Address Space of the CPS server through extraction of the BOM from CAD or migration from legacy DB to enhance the functionality of the physical engine based on the BOM. Simulator advancements can be achieved with physical engine and Address Space. For the basic process of design management, all changes to the design for each stage are recorded in the DB, and then the rendering server lists the version history by referring to the DB. A blueprint for each stage of each VR/AR version can be recalled. It is possible to manage the version of each design level and real-time management of changes to each VR/R-based design. Products produced through VR collaboration design can be inspected using AR technology. When selecting a part list, the presence or absence of the relevant part, specifications, etc., are displayed to check for abnormalities.

Physical engines can simulate behavior without manufacturing products by applying optimized physical engines of certain real-world devices. If important design modifications—such as those involving parts, materials, and appearances—are made by sending data to CPS in real-time simulation, the appropriateness or stability of the change can be verified.

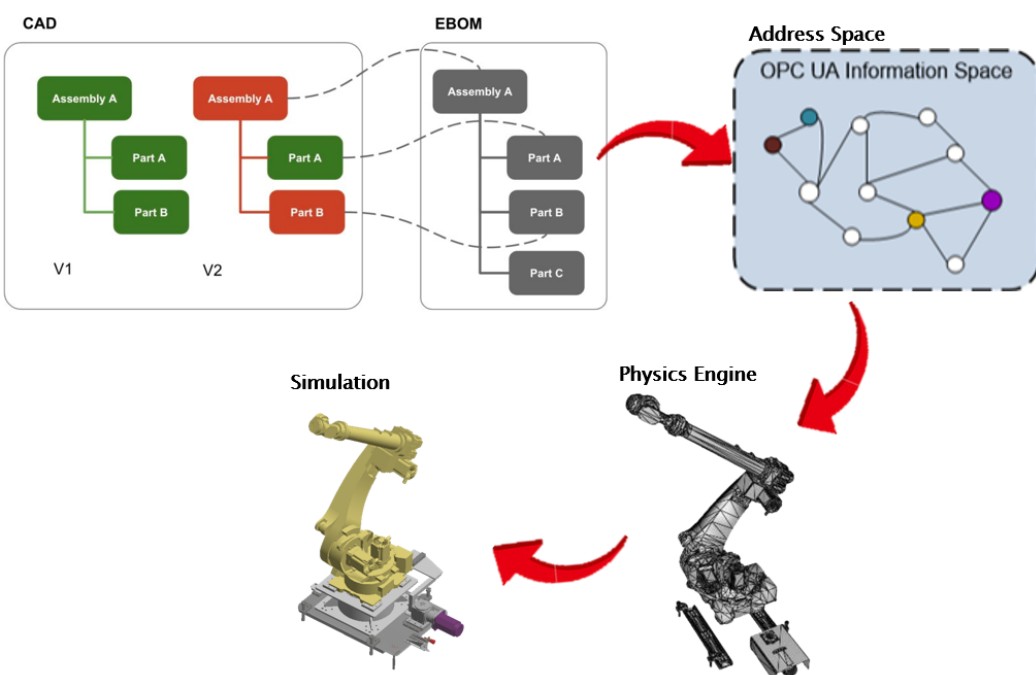

**Figure 6.** Simulation of CPS interworking.

## 4. Implementation and Results

### 4.1. System Configuration

In this paper, OPC UA Foundation's SDK and C language were used. The basic functions of CPS server and Smart Connect were implemented as shown in Table 1 using the source of the CPS server part of Flexing Server [29], which developed the CPS server. Smart Connect is part of the CPS server, and each distributed system is implemented as a CPS server and OPC UA server protocol that can have a centralized address space.

**Table 1.** Implementation of CPS Server.

| Classification | CPS Server |
|---|---|
| Operating system | Windows 10 Pro |
| Specification | AMD Ryzen9 3940X 16-Core, Mem 128 G, SSD 256 GB |
| Purpose | Receive BOM data from VR Parser and configure Address Space on CPS Server in real time |

The Smart Connector needs to generate NodesetXML to be applied to the CPS Server based on data received from the Socket (.iam file to open node). It reads element ID information from an XML file, collects relevant BOM information, and generates objects from the model. CAD Socket Class—serves as a socket server that receives data from the VR Parser. Figure 7 shows the details of the process. When VR Parser converts the .iam file parsed by the Socket Client into the Json format and sends the data to the CAD Socket, tcpListenerThread sends the received data to the NodeSetXmlGenerator after checking the received data. TCP Listener was implemented with Newtonsoft.json NuGet package with C-based NetCore 3.1 version, and NodeSetXML file generation was studied according to the OPC UA standard.

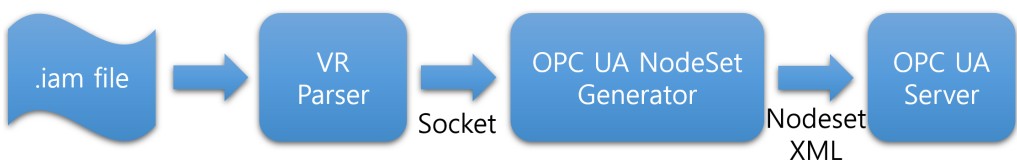

**Figure 7.** Smart connector structure.

It is possible to change the OPC Node in real time, collect process device data, and change the .iam file data. Finally, we study custom scripts and VR Parsers to parse the corresponding Json file and extract relevant information before re-generating the model by implementing the C class in the Unity SDK. Table 2 details the implementation environment.

**Table 2.** VR Parser Classification.

| Classification | VR Parser |
|---|---|
| Operating system | Windows 10 Pro |
| Specification | AMD Ryzen 3 3100 4-Core, Mem 32 GB, SSD 256 GB |
| Purpose | Purpose of extracting BOM data from CAD files |

To parse the equipment or CAD File data into OPC UA, VR Parser is UI-verifiable and visualizes 3D models. Users can browse models; search among engineering models, specifications, and configurations in a VR environment; check them through the UI as shown in Figure 8; and change the OPC Node in real-time and process the device data.

**Figure 8.** Smart connector and VR parser.

Moreover, high-spec servers are required for 3D model rendering, and the server configuration environment is as described in Table 3. There are performance limitations to performing rendering of 3D models on user devices, and 3D models are generated and streamed to devices through predefined 3D modeling information. This allows for the designed product to be simulated by utilizing the AR function provided by Vuforia Engine, a physical engine, as if it were operating in a real environment. The Vuforia EnginE includes

image recognition, QR recognition, and AR object viewer functions, which are key AR technologies, and it is optimized for Unity engines.

**Table 3.** Implementation of Rendering Server.

| Classification | Rendering Server |
|---|---|
| Operating system | Windows 10 Pro |
| Specification | AMD Ryzen 9 3900X, Mem 64GB, SSD 4TB |
| Purpose | 3D model rendering and streaming |

*4.2. Implementation*

Company T, located in Pyeongtaek, Korea, manufactures assembly line automation facilities and specialized equipment (piston insertion robots, valve finishing devices). In fact, there are several facilities and various robots available for use, but in this study, as shown in Figure 9, one 7-axis robot arm was used to model a physical factory environment.

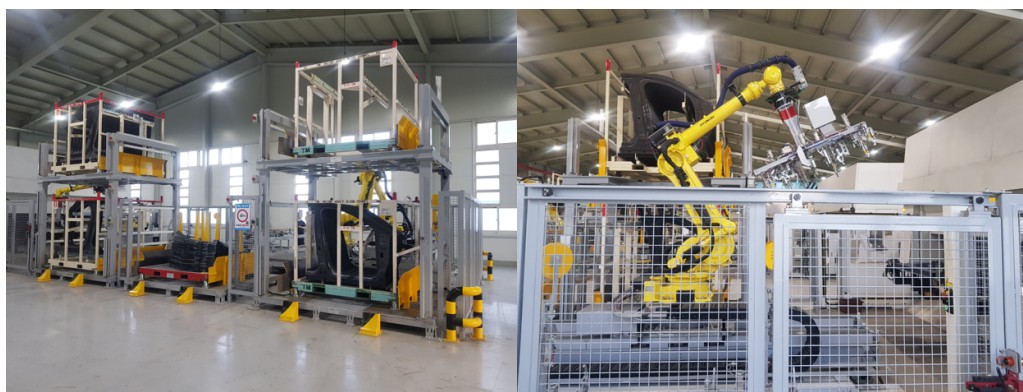

**Figure 9.** Automated facilities of Company T's.

Data collection was required for designing the 3D VR production of the robot arm, and a 2D information product drawing, automated mechanical equipment 3D CAD information, and photo base mapping were conducted. Modeling data, including layers of components of each automated mechanical device and layers of each component, were collected. In the drawing of the acquired automation machine, the storage extension conversion was carried out through an inverter, and the layer was parted in the design program to form a part-oriented layer. Upon completion of the extension conversion, it was loaded into a real-time engine after being converted to general data such as IGS, OBJ, and 3DS. Further, by applying the Surface Name, a preparation step for material and baking was prepared. After defining the material of the robot arm based on an image, VR content sets on the same scale as actual equipment are configured, and graphic performances such as unnecessary polygons and high-calculated material improvement are optimized. CAD design files of real-world physical automation machines are converted to 3D objects. CAD design files exist in various formats, but they are basically conducted based on commonly used inventory. As shown in Figure 10, the 3D CAD design file for the automation machine includes data on the BOM and the 3D shape.

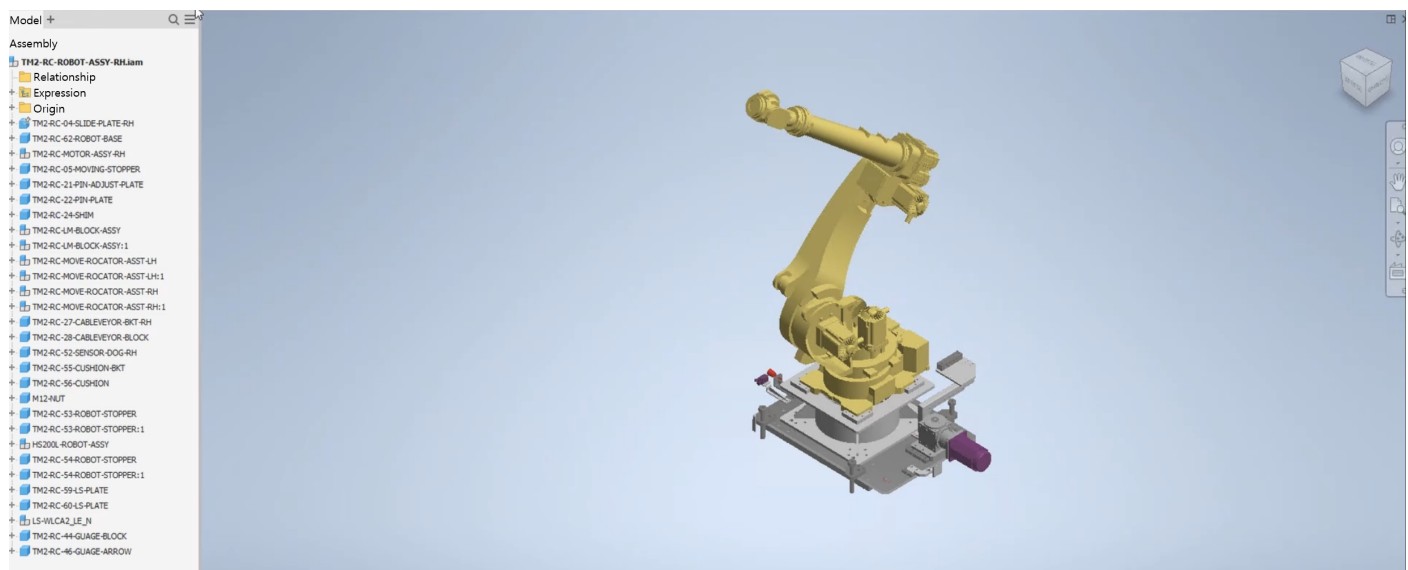

**Figure 10.** .iam Design File with Inventor.

The implemented VR engineering process was implemented in a total of four projects as shown in Figure 11. FBXExport wraps it and delivers it in DLL format to use FBX-related API in Unity3D project. CAD Convertor creates and displays a mesh on the screen based on DLLs such as CAD and FBX, and delivers it to the cloud server. VR Application uses libaray such as AR Core, Intelloid, and WebRTC to build a platform in AR environment and deliver it to users. The VR Projects Parser uses VB.net to transmit a program written through an actual Inventor to CAD Convertor through socket communication. The reason why it is written in VB.net is that the API provided by Inventor is provided through VB.net, and it is transmitted as a loopback for sharing between processes.

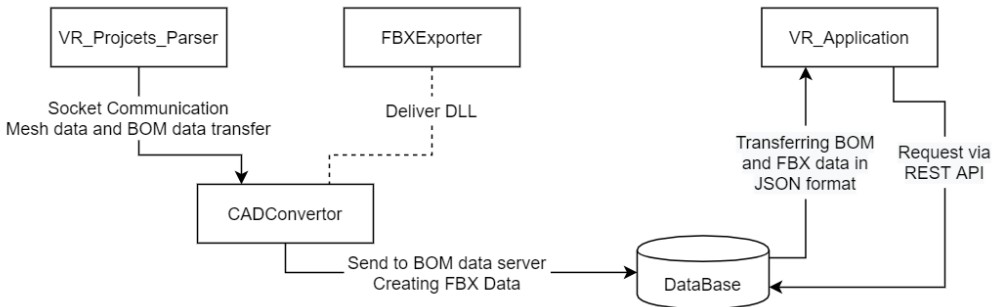

**Figure 11.** VR engineering project configuration.

A CAD Converter is a necessary tool for converting these 3D CAD design files into 3D objects in a VR environment. The CAD Converter role is performed by the VR Parser developed in this study. The .iam file should be loaded as shown in Figure 12 to read the information in the CAD file. When all are loaded, BOM and 3D objects are created.

By implementing the BOM data and metadata in CAD design files as the Address Space of CPS, a VR Parser can transmit BOM data to an OPC UA-based CPS server. When the transmission is completed, the OPC UA client program can access the CPS server to verify that the structure of the BOM data is created in the Address Space. Further, when accessing the CPS server, it can be confirmed that the structure of the BOM data is generated as shown in the figure. As shown in Figure 13, the BOM structure of the CAD file was typically generated as the Address Space in CPS, and the entire part information and the operable part information were implemented in the Address Space.

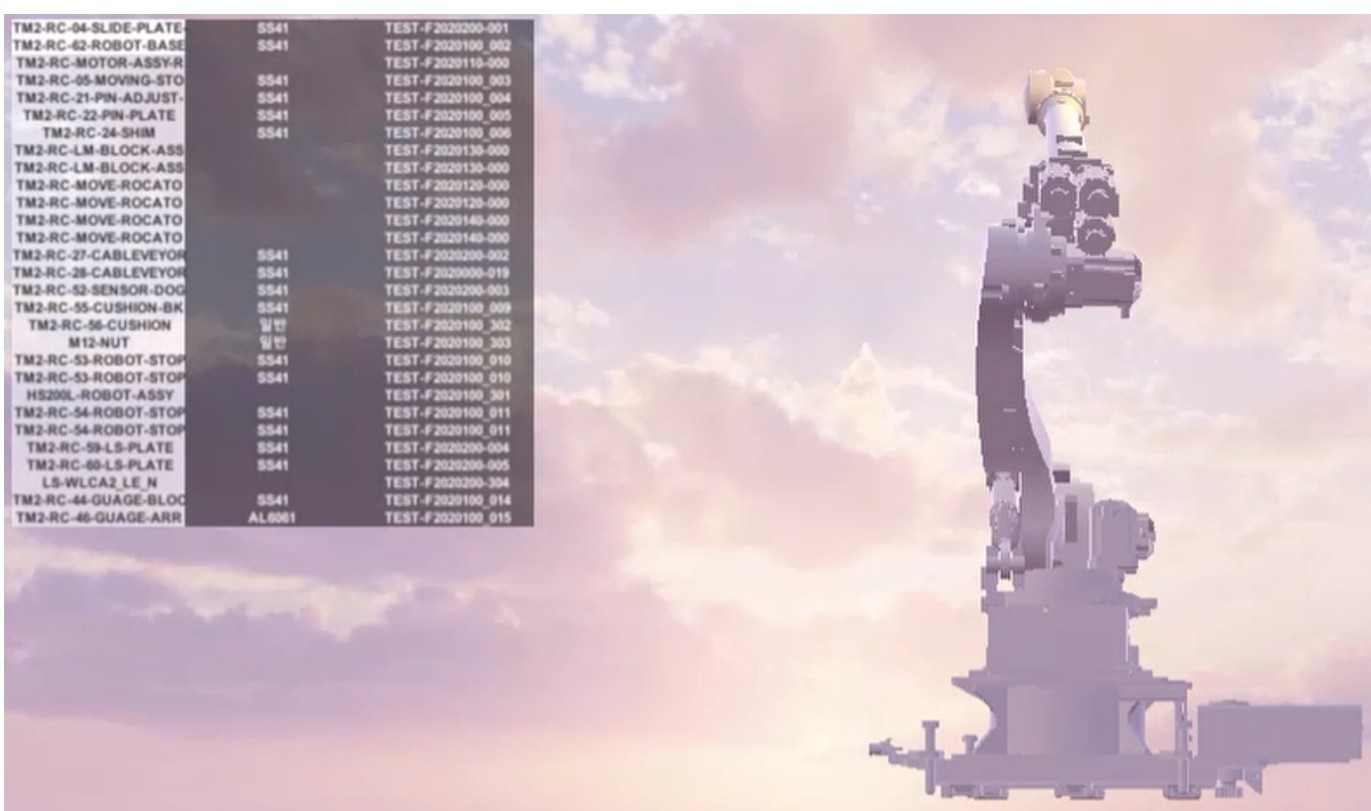

**Figure 12.** Creating a 3D object in a VR Parser.

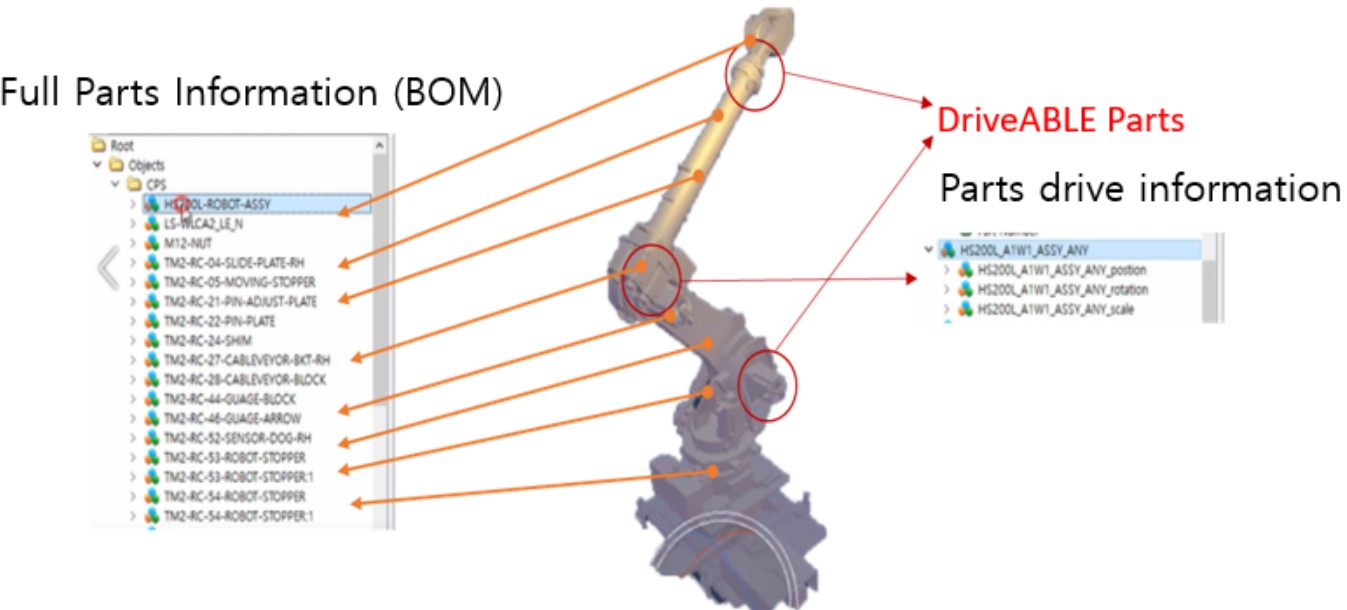

**Figure 13.** Address Space implementation of 3D objects.

To optimize the FBX model file capacity, the CAD converter can extract 3D objects as FBX models. When EXPORT is performed, it is automatically extracted as an FBX model. The capacity is optimized for extraction and default is 50%. In Figure 14, it can be observed that the FBX 3D model is made normally.

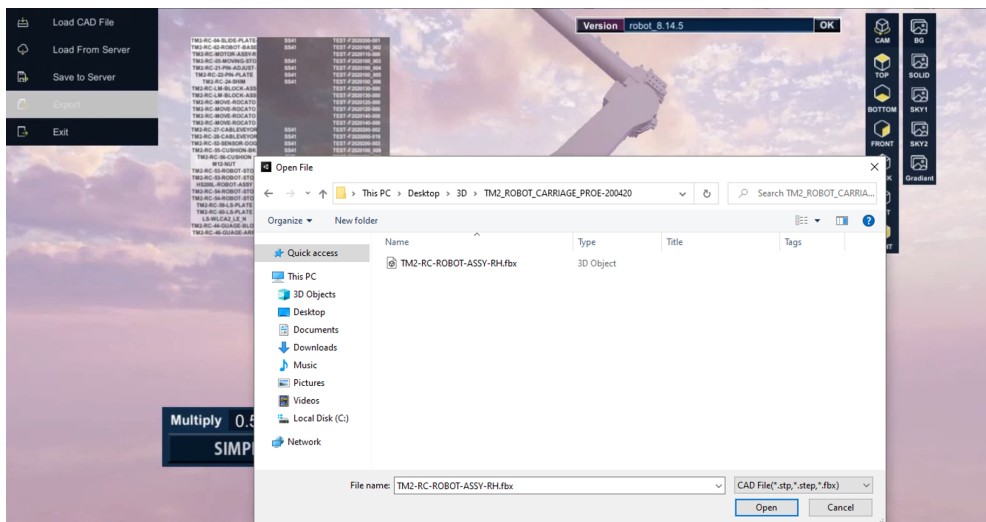

**Figure 14.** FBX model file capacity optimization.

In the CAD file, the robot arm was simulated by adding a physical engine to confirm the physical force of the model converted into a 3D object. For the simulation, a physical engine that calculates the physical force for the load through two robot arm colliders has been added. The collider is a component for collision detection and can add physical functions with scripts. Unity Physics and Havok physics were added to each automated machine, as shown in Figure 15.

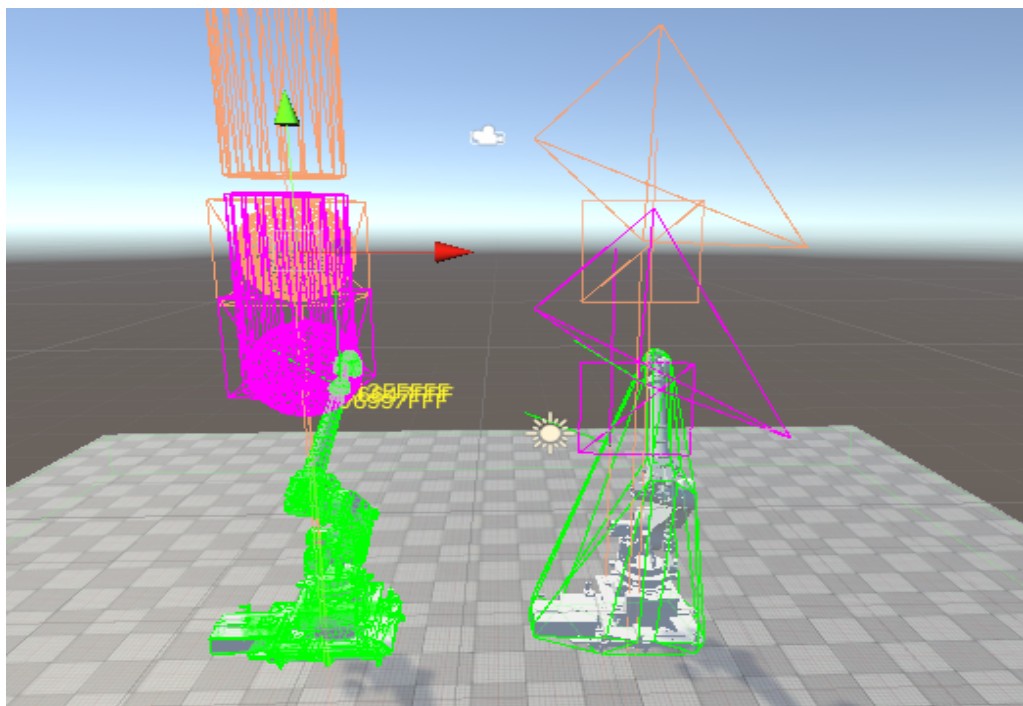

**Figure 15.** Physical engine simulation of unity physics and Havok physics.

After importing the designed .iam file into the VR Parser, the design problem is checked through load simulation, and the stress–strain rates of specific parts due to the action of vertical stress resulting from the load in the size and shape, and the structure of the designed file is checked as well. It expresses the collider of each other object and expresses the movable range of the object. When a condition exceeding the yield point on the diagram occurs, the problematic part is expressed through the VR Parser, and the user can check the wrongly designed part through this simulation.

*4.3. Results*

BOM data is extracted from CAD files created by inventory through VR Parser, converted to Json, and delivered to the CPS Server. At this time, the BOM data of the CAD file modified in the inventory is well transmitted to the CPS server, and the BOM data from the VR Parser checks the accuracy of the configuration of the address space of the CPS Server in real time. Further, the FBX file extraction function for external program interworking compresses the data in the CAD file to measure the capacity optimization (capacity reduction) value. The test environment was constructed as depicted in Figure 16.

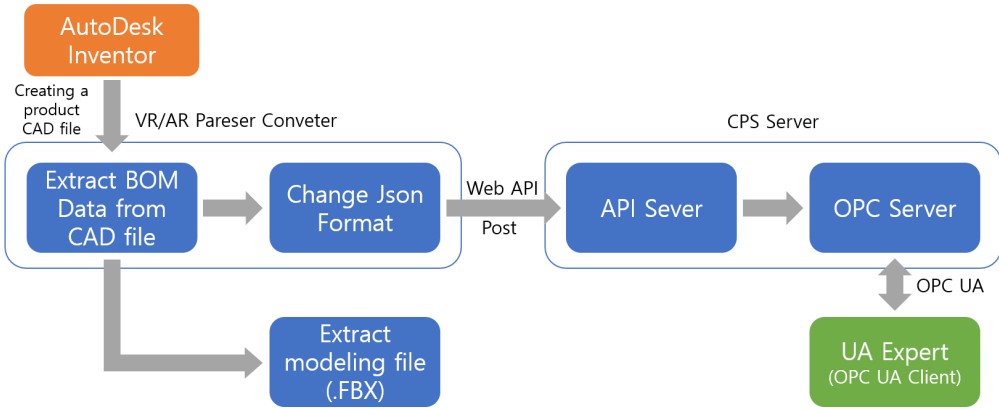

**Figure 16.** Experiment diagram.

The total number of CAD files to be converted is 5, and the total number of data objects is 77. After running the inventory, the CAD file should be opened to check the CAD BOM data. The VR Parser should then be run to load CAD files. As shown in Figure 17, UA Expert is a representative client that supports the OPC UA protocol with software from a company called Unified Automation. The BOM data of the inventory, the number of nodes converted to OPC UA, Node name, Node tree structure, and Node properties can be verified by running the most commonly used internationally test client that supports OPC UA functions such as data access, alarms, history, and UA method calls.

This procedure is executed by the number of CAD files, and the accuracy converted to the expression of (1) is calculated.

$$\text{Conversion accuracy} = \frac{\text{Number of Objects Converted to Address Space}}{\text{CAD File Total Objects}} \times 100 \qquad (1)$$

As a result of the data conversion test, the accuracy of address space conversion in BOM was 100%, as shown in Table 4, thus indicating the reliability of BOM product inspection. This shows that real and bidirectional configuration results are visualized in real time in collaborative design and that interaction between users is possible.

**Table 4.** Translation accuracy of Address Space in BOM.

| Number | CAD File Total Objects | Number of Objects Converted to Address Space | Results |
|---|---|---|---|
| 1 | 29 | 29 | |
| 2 | 28 | 28 | |
| 3 | 3 | 3 | 100% |
| 4 | 14 | 14 | |
| 5 | 3 | 3 | |
| Sum | 77 | 77 | - |

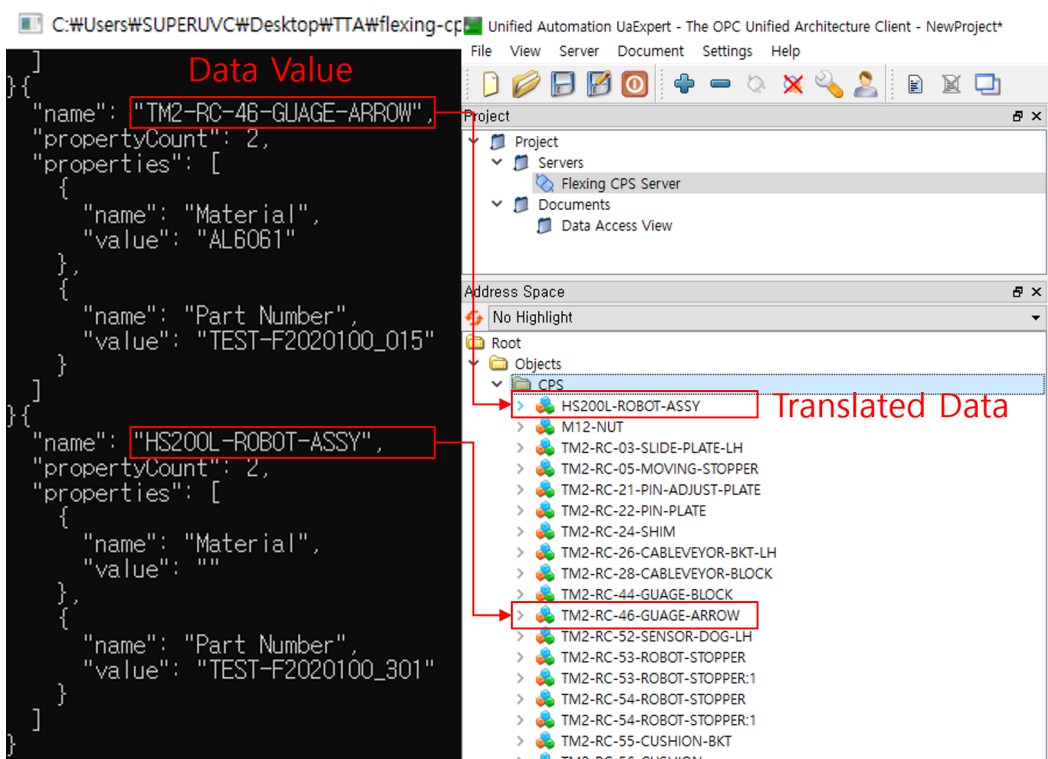

**Figure 17.** Data conversion of CPS server.

For CAD data, it should be converted to a data format that can be processed by the VR engine. The Unity engine can only handle two types of data: FBX and OBJ. Almost all CAD systems have data preprocessing for generating FBX and OBJ data, but the generated FBX and OBJ do not have the quality to be used directly as VR data. The FBX and OBJ data formats visualize the 3D shape of the object using polygon elements, and the number of polygons used is critical to the 3D graphic quality of the visualization object. The higher the number of polygons, the more realistic is the 3D object and the 3D graphics quality is higher. However, having a large number of polygons increases data volume and leads to performance problems in VR applications. Since this study requires optimization to ensure a balance between graphic data quality and VR application performance, the optimization value is calculated by calculating the capacity size of the extracted FBX file and the existing CAD file using Equation (2) by importing and exporting CAD files from the VR Parser. Figure 18 shows the optimization results.

$$\text{Optimization Value} = \frac{\text{File Size After Optimization}}{\text{CAD File Size}} \times 100 \tag{2}$$

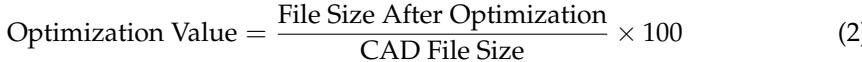

| Name | Date modified | Type | Size |
|------|---------------|------|------|
| Mesh1.fbx | 2020-12-16 오후 2:35 | 3D Object | 1,036KB |
| Mesh2.fbx | 2020-12-16 오후 2:35 | 3D Object | 1,005KB |
| Mesh3.fbx | 2020-12-16 오후 2:35 | 3D Object | 408KB |
| Mesh4.fbx | 2020-12-16 오후 2:35 | 3D Object | 125KB |
| Mesh5.fbx | 2020-12-16 오후 2:35 | 3D Object | 148KB |

**Figure 18.** Data optimization size.

The results of the optimization test comparing the size of the CAD design file, as presented in Table 5, show an average optimization value of 37.4%. We confirmed an improvement of 12.6% over the optimization default of 50%, which can be expected to

improve the performance of the same VR application and the quality of 3D graphics in the IT infrastructure.

**Table 5.** CAD Design File Optimization (Reduction).

| Number | CAD File Size (KB) | File Size after Optimization (KB) | Results |
|---|---|---|---|
| 1 | 2004 | 1036 | 51.7% |
| 2 | 2214 | 1005 | 47.3% |
| 3 | 1103 | 408 | 37.0% |
| 4 | 790 | 125 | 15.8% |
| 5 | 421 | 148 | 35.2% |
| Sum | - | - | 37.4% |

To measure the graphic quality through optimization, the influence and quality of rendering were tested. It should be verified that real-time VR rendering content can be provided by importing CAD files from VR Parser and then exporting them to convert the extracted FBX files into 3D formats. The test was conducted as shown in Figure 19, while focusing on the frame rendering speed and the resolution of the 3D content.

Based on the 3D format-converted 2D files, 3.5.99 frames per second was used to measure the number of frames per second of VR image rendered through CAD converter, and the number of frames per second was checked after moving to the left, right, and top of VR images. Further, the resolution of the rendered VR image was confirmed.

Rendering to CAD files in a VR system can result in high transmission load, slow rendering speed, and poor 3D control due to the generation of large amounts of polygons, so it should be verified that it is efficient for VR systems. In existing studies [47] aiming to simplify CAD files, it was measured at 75 frames when optimization was performed at 50%. Converting CAD data into FBX, in most cases, decomposing a single object into a sub-object, damaging the object structure, which was feared to degrade performance, but the frame measurement results of the model provided as VR Parser are shown in Table 6. Full view from parent of 1, the overall foreground was 86 frames per second, Full foreground from left end 2, the preliminary foreground was 88 frames per second, and Full foreground from right end of 3, the overall foreground was 83 frames, showing an average result of 85.6 frames. It also provides a high resolution of 2K for resolution.

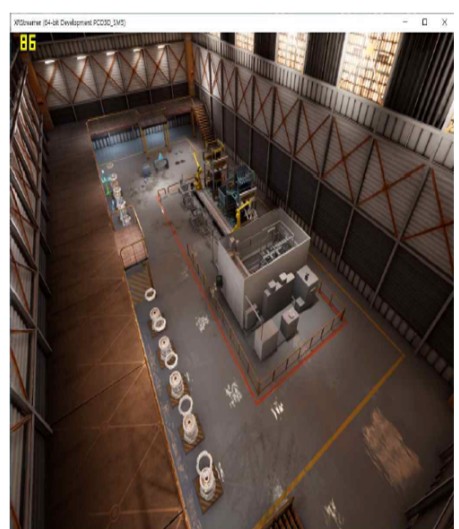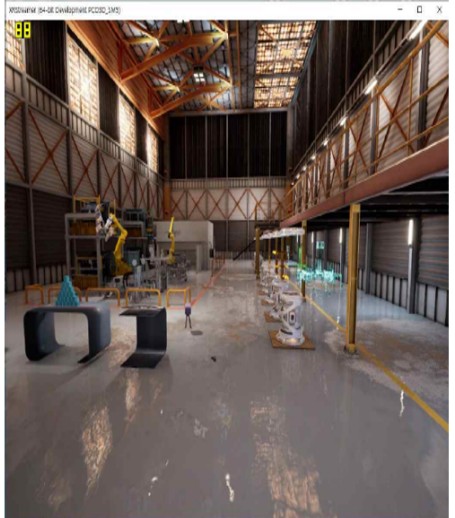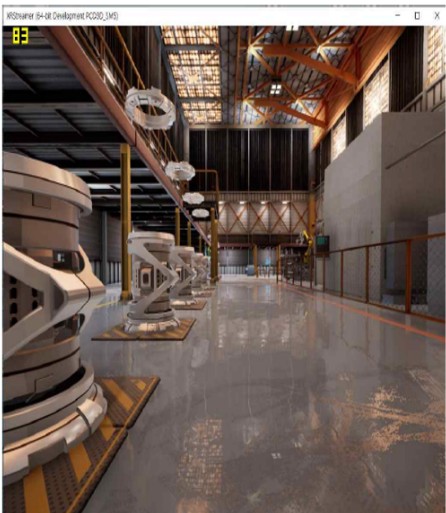

**Figure 19.** Frame rendering speed.

**Table 6.** Frame count and resolution per second.

| Classification | Measurement Location | Frames per Second | Resolution |
|:---:|:---:|:---:|:---:|
| 1 | Full view from parent | 86 | 2K (2048 × 1080) |
| 2 | Full foreground from left end | 88 | 2K (2048 × 1080) |
| 3 | Full foreground from right end | 83 | 2K (2048 × 1080) |
| Average Number of Frames per Second | - | 85.6 | 2K (2048 × 1080) |

## 5. Conclusions

In this paper, we proposed a VR/AR collaboration model and VR engineering process using CPS applied with international standard technology. Using VR/AR rendering of product design 3D drawings in the manufacturing industry, a collaborative model was presented to allow participants to participate in design at the same time, and an application was developed to check the virtualization model with data collected through OPC UA-based CPS. The programs used in machine design and product design currently use 100% foreign software. However, the current foreign software is specialized in design, but collaborative design does not provide functionality, so the actual design process of reviewing and reflecting opinions, and then simulating them, can be replaced with domestic programs to cover the growth of foreign design programs. Further, domestic manufacturers will find problems later by starting products or main production after design is completed, and it will take a long time and high cost due to the method of re-reflecting and re-production. However, sufficient simulation and design review using the product inspection system can dramatically reduce this time and cost, thus contributing to the competitiveness of manufacturing companies and actively using it for smart factories in the manufacturing industry.

In this study, we developed a converter that converts design CAD files and BOMs into graphic models and CPS models, and that successfully loads BOM data of 3D CAD files from the VR Parser and sends BOM data to CPS server to construct Address Space. We also successfully extracted 3D objects converted from CAD files with capacity-optimized FBX. A CPS server, a middleware operating under the OPC UA protocol, was established to link OPC UA–BOM Address Space with OPC UA–BOM-based real-time values. In addition, when converting to 3D VR content using CAD drawings, it was implemented in the same way as the measured equipment, and errors were determined in CPS through the calculated load value of the load in the physical engine as the value generated by the simulator.

Here, time and competency were both insufficient to implement all of the proposed VR/AR collaboration models. In the future, we will develop a VR/AR collaborative design client to create virtual characters for multi-user access, VR and motion capture wearers, and build a WebRTC-based backend (server and service) to stream high-end content and voice communication between users using VoIP.

**Author Contributions:** Conceptualization, J.K.; methodology, J.K.; validation, J.K. and J.J.; formal analysis J.K.; investigation, J.K.; resources, J.K.; data curation, J.K.; writing—original draft preparation, J.K.; writing—review and editing, J.K.; supervision, J.J.; project administration, J.K.; funding acquisition, J.J. All authors have read and agreed to the published version of the manuscript.

**Funding:** This research was supported by the MSIT (Ministry of Science and ICT), Korea, under the ITRC (Information Technology Research Center) support program (IITP-2022-2018-0-01417) supervised by the IITP (Institute for Information & Communications Technology Planning & Evaluation). Also, this work was supported by the National Research Foundation of Korea (NRF) grant funded by the Korea government (MSIT) (No. 2021R1F1A1060054).

**Institutional Review Board Statement:** Not applicable.

**Informed Consent Statement:** Not applicable.

**Data Availability Statement:** Not applicable.

**Acknowledgments:** This research was supported by the MSIT (Ministry of Science and ICT), Korea, under the ICT Creative Consilience Program (IITP-2022-2020-0-01821) and the ITRC (Information Technology Research Center) support program (IITP-2022-2018-0-01417) supervised by the IITP (Institute for Information & communications Technology Planning & Evaluation).

**Conflicts of Interest:** The authors declare no conflict of interest.

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
