# Peer review of "Design and Implementation of OPC UA-Based VR/AR Collaboration Model Using CPS Server for VR Engineering Process"

_applsci, doi:10.3390/app12157534_

Round 1

Reviewer 1 Report

This paper is interesting and addresses an active topic. However, it was not written quite professionally. After reading the paper, I have the following comments:

1. The proposed solution is well described in the introduction. However, what is missing is the identification of the end users or for whom the solution is intended. Identify more whether the solution is suitable for large enterprises ready for computationally intensive operations or whether it is also suitable for small and medium-sized enterprises. A link may provide some inspiration for incorporating comments above:
https://susy.mdpi.com/user/manuscripts/review_info/434a221563b8abd2d47c8cdf21669ffa

2. In Chapter 2, authors claim they "developed a hardware and software-based user interface". In the following chapters, the hardware solution is only vaguely outlined. Can you please clarify this?

3. It would be of great help for the paper to describe the model in a suitable modelling language such as UML. Instead, was used an image that does not convey the complexity of the solution. Please include the model in standard form.

4. A software solution developed by authors was also mentioned. However, it is not clearly defined in the article or in block notation, nor are provided code samples.

In my opinion, this is an interesting topic. However, the article needs to rework the bland parts and focus more on its own contribution to the issue addressed.

General comments:

1. Please check text for typos and mistakes.

2. Are all the figures your own work?

3. Most of the figures are blurry or otherwise inappropriate for publishing:

Figure 5 - please correct figure caption

Figure 7 - very blurry, can hardly read the labels

Figure 9. - needs clear explanation, very hard to read. I recommend listing the terminal output as text, not a screenshot.

Figure 10. - very blurry. What should it tell to the reader?

Figure 11. - what is "Inventory" in figure caption?

Figure 12. - unreadable!

Figure 17. - very hard to read. I recommend listing the terminal output as text, not a screenshot.

Reviewer 2 Report

A  design review process for the cyber-physical system by utilising the Open Platform Communications Unified Architecture is proposed in this work. The system integrates  AR/VR to OPC UA, allowing the CPS design to be tested in a virtual environment. The first part of the paper explains the system architecture and the second part provides the implementation results showing UI screenshots of each sub-module. The topic discussed is of relevance to the manufacturing sector. However, the article is not suitable for publication in its current form. A thorough review is recommended. Following are some of the points that need attention.

1.       The author may minimise the abbreviations in the title and the abstract. Many times, expansion was not mentioned at the first use, nor was the list of acronyms given.

2.       It is recommended to split the long and complex sentences into simpler ones. Many times, it is difficult for the reader to understand the point being discussed by the author. The whole paper needs a thorough revision in this regard.

Some examples are:

"The converting program that converts CAD/BOM into 3D graphics model and CPS model, and CPS server, an OPC UA-based middleware are implemented to provide 3D modeling optimisation, and the system is tested through robot arm data in the actual factory and its performance is studied in the laboratory."

"Such manufacturing can communicate and interact with the environment and other products using IoT technology and web-based services."

" With the development of the technology of the 4th industrial revolution, the smart factory market of the manufacturing process is expanding "

3.       In order for manufacturers to overcome the COVID-19 pandemic crisis, it is pointed out that it is necessary to actively introduce robot automation that can promote safety of employees and increase production efficiency, and to combine virtual convergence reality technologies such as digital twin (Virtual Reality)/AR (Virtual Reality). Is digital twin and AR the same as VR? Are you referring to digital shadow? you may refer following papers for clarity

a)          Li, Qian, et al. "Design and implementation of interactive digital shadow simulation system." Communication Systems and Information Technology. Springer, Berlin, Heidelberg, 2011. 187-194.

b)         Ramasubramanian, Aswin K., et al. "Digital Twin for Human–Robot Collaboration in Manufacturing: Review and Outlook." Applied Sciences 12.10 (2022): 4811.

c)          Barricelli, Barbara Rita, Elena Casiraghi, and Daniela Fogli. "A survey on digital twin: Definitions, characteristics, applications, and design implications." IEEE access 7 (2019): 167653-167671.

d)         Zheng, Yu, Sen Yang, and Huanchong Cheng. "An application framework of digital twin and its case study." Journal of Ambient Intelligence and Humanized Computing 10.3 (2019): 1141-1153.

4.       The robots are introduced to increase the safety of the operator during the pandemic. What will happen to the operator and the robot once the pandemic is over?

5.       The last paragraph of the introduction section that explains each section of the paper should be revised.

6.       How is the current model different from the other similar models used for manufacturing collaboration?

7.       Page 3 Line 33 states that "As the switch to CPS is made, more data will be generated in the future, improving the physical environment". How the physical environment improves when the data is gathered?

8.       Is complete metadata, including material and tolerance information of parts necessary for Engineering applications of  AR/VR? I can see that projects like https://www.sherlock-project.eu/home use AR/VR for assembly operations.

9.       Figure 1 is copied from reference 23. It is assumed that permission is taken to use the figure in the current paper. Proper credit should be given for the same.

10.   The intext explanation of figure 4 is not sufficient enough to understand the related concept. Further, it seems that the image shown in the centre (Real video stream) is stitched onto the image of the tablet and the computer located in front of the person. Does Figure 4 show the system developed in the present work? Similarly, in Figure 7, the part in the simulation doesn't seem to be the same as the part shown in the cad model.

11.   The quality of all the figures should be improved. It is hard to read the printed text.

12.   Table 1, 2 and the intext explanation could be improved.

13.   Figure 10 is not clear. Modelling the robot might not be enough for modelling the physical factory. What about the model of the environment?

14.   "In this paper, the source of OPC UA Server part of Flex Server [23], OPC UA Server, developed OPC UA Server using OPC UA Foundation's SDK and C language, used the basic functions of OPC UA Server and Smart Connect part of the OPC UA Server part, and implemented each distributed OPC UA server or AddPS server protocol." The authors have failed to communicate what their contribution is. Is this work merely a proposal of new architecture? Or Is this architecture implemented and tested?

15.   The use of two physics engines in the simulation should be explained?

16.   How does software originating outside a particular geographical region behave differently in another region unless special restrictions are made? How does the origin of the software affect its functionality? It is observed that the current method also utilises software originating from a different geographical region. Proper Justification should be given. There are many more examples similar to the above example where proper Justification is necessary.

Reviewer 3 Report

The paper is of interesting and up-to-date topic and I find it as well-written and worth of publication. In fact I have only few minor remarks/suggestions that can help to improve the manuscript:

1.       I suggest to add the goal of the paper and information of most important result in the abstract. On the other hand it is not necessary to put information about development of CPS technology in Korea (therefore I suggest to remove the sentences from lines 6-13 or rewrite them to make them more general – i.e. not focused on Korea).

2.       In introduction you describe Digital Twins. I suggest to add some examples of usability of this technology in industrial environment (see e.g. papers with dois: 10.1016/j.cie.2022.108262, 10.3390/app12115727, https://doi.org/10.3390/s21237830, 10.1155/2022/4879490, 10.1007/978-3-031-00805-4_4, 10.1016/j.ifacol.2022.04.209, etc.)

3.       I suggest to explain more clearly how the VR/AR technology can be used together with digital twin and what are the advantages of such a solution in industrial conditions.

Round 2

Reviewer 1 Report

The paper is not scientific enough, although due to the hard work you put into it, I accept it.

Reviewer 2 Report

The authors have addressed most of the comments and proofread the article with the help of a professional English language document review and editing centre. The issue of plagiarism is also addressed.

11. Minor grammatical/spelling mistakes are spotted (Eg. Line 126, 133, 247 etc), acronym- 121.

Line 247, for example, is grammatically correct; however, the point made is not right. I think virtual reality refers to an artificial environment that feels like reality rather than artificially creating an environment that feels like reality. Similar mistakes are spotted along the paper. Authors may review the whole article to address such errors.

22. Previous work, some of them listed below, shows the use of OPC UA to create DT/ VR/AR of CPS. Hence, it is recommended to add a subsection or paragraph explaining the difference between such existing systems and the current one.

a.       Model-Based Interconnection of Digital and Physical Twins Using OPC UA

b.       Improving interoperability of Virtual Commissioning toolchains by using OPC-UA-based technologies

c.       Human-Robot Interaction via a Virtual Twin and OPC UA

33. Figure 4 may further be improved by avoiding copying the central image onto the phone and the computer screen on the right side. Does the picture( computer) shown at the right bottom need reference? Or is it made by yourself?
